# Topological optical differentiator

Tengfeng Zhu[1,2], Cheng Guo [1], Junyi Huang[2], Haiwen Wang[1], Meir Orenstein[3], Zhichao Ruan [2✉] & Shanhui Fan [1✉]

Optical computing holds significant promise of information processing with ultrahigh speed and low power consumption. Recent developments in nanophotonic structures have generated renewed interests due to the prospects of performing analog optical computing with compact devices. As one prominent example, spatial differentiation has been demonstrated with nanophotonic structures and directly applied for edge detection in image processing. However, broadband isotropic two-dimensional differentiation, which is required in most imaging processing applications, has not been experimentally demonstrated yet. Here, we establish a connection between two-dimensional optical spatial differentiation and a nontrivial topological charge in the optical transfer function. Based on this connection, we experimentally demonstrate an isotropic two-dimensional differentiation with a broad spectral bandwidth, by using the simplest photonic device, i.e. a single unpatterned interface. Our work indicates that exploiting concepts from topological photonics can lead to new opportunities in optical computing.

[1] Department of Electrical Engineering, Ginzton Laboratory, Stanford University, Stanford, CA, USA. [2] Interdisciplinary Center for Quantum Information, State Key Laboratory of Modern Optical Instrumentation, and Zhejiang Province Key Laboratory of Quantum Technology and Device, Department of Physics, Zhejiang University, Hangzhou, China. [3] Department of Electrical Engineering, Technion-Israel Institute of Technology, Haifa, Israel. ✉email: zhichao@zju.edu.cn; shanhui@stanford.edu

Optical computing, which exploits the propagation of light for computing purposes, has been of interests for many decades due to the potential of performing computation at high speed and low power consumption[1,2]. In particular, Fourier optics setups have been widely explored for performing many of the computations that are useful for image processing[3–6]. These setups, however, are generally rather bulky. In recent years, there is a renewed interest in optical computing thanks to the developments of on-chip silicon photonic circuits and nanophotonic structures, which enable optical computing with devices that are far more compact compared with their conventional counterparts[7–12].

A prominent example of optical computing is the spatial differentiation operation, which is useful for edge detection in image processing[13]. There have been substantial efforts in the past several years on the theoretical design of various nanophotonic structures for spatial differential operation in either one[14–24] or two dimensions[25–28]. Most of the experimental demonstrations, however, are restricted to one-dimensional differentiation[29–37]. There are two recent reports of two-dimensional differentiation using a photonic crystal structure[38] and a multilayer absorber[39]. However, the photonic crystal differentiator does not provide an isotropic operation, since doing so requires an elaborated design of the structure[25]. Furthermore, both of the methods operate only in a relatively narrow spectral bandwidth, since they both rely upon the use of optical resonances.

In this paper, we show that some of the concepts in the field of topological photonics can be applied in optical computing for the purpose of achieving broadband isotropic two-dimensional differentiation. To achieve two-dimensional differentiation, by the definition of differential operator, the transfer function of the device should exhibit an isolated zero in the wavevector space. Since the transfer function is in general complex, such an isolated zero should generically carry a topological charge[40]. Here we show that such a topological charge of ±1 can be straightforwardly achieved in a cross-polarization setup of reflection at a dielectric interface. Moreover, an isotropic transfer function with such a topological charge can be generated by adjusting experimental parameters. As a result, we provide an experimental realization of isotropic two-dimensional differentiation over a broad spectral band based on mere reflection from a flat surface. Different from the work which uses topological states for one-dimensional transfer functions[11], we directly construct a topological charge in two-dimensional transfer functions to engineer the functionality of device.

## Results

**Theory**. Our setup is based on reflecting an incident polarized beam from a flat surface and subsequently analyzing the reflected beam with a polarizer, as schematically shown in Fig. 1. We define the beam profile in terms of coordinates $x$ and $y$ that are perpendicular to the beam propagation direction, with axes $x$ and $y$ being parallel and orthogonal to the plane of incidence, respectively. For an incident (reflected) plane wave propagating along the beam propagation direction, its electric field components along the $x$ ($-x$) and $y$ directions correspond to the $p$ and $s$ polarizations, respectively. In the paraxial regime, the incident beam has the form $\mathbf{e_{in}}S_{in}(x,y)$, where a 2-vector $\mathbf{e_{in}}$ in the $x$–$y$ plane describes the input polarization, and $S_{in}(x,y)$ describes the scalar electric field distribution on the plane perpendicular to the beam propagation direction. By the spatial Fourier transform, the incident field can be decomposed into spatial frequency components through

$$S_{in}(x,y) = \iint \tilde{S}_{in}(k_x, k_y) \exp(ik_x x) \exp(ik_y y) dk_x dk_y, \quad (1)$$

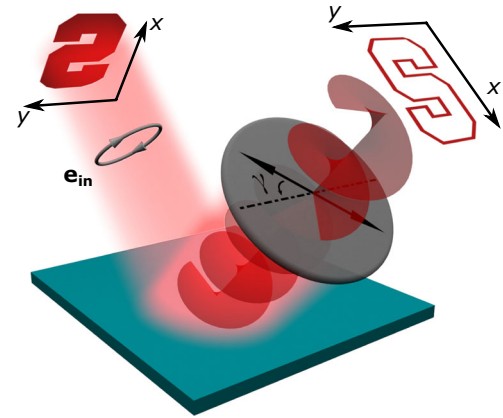

**Fig. 1 Schematic of the experimental setup, where an incident beam with polarization $\mathbf{e_{in}}$ is reflected by an interface, and then passes through a polarizer which selects a linear polarization with an orientation angle $\gamma$.** This setup performs an isotropic two-dimensional spatial differentiation, and can generate an output beam with an orbital angular momentum of ±1 with an input Gaussian beam. $x$ and $y$ are the beam profile coordinates for the incident and reflected light, which share the same origin on the interface.

where all wavevector components have the same polarization $\mathbf{e_{in}}$. Because of the continuous condition of the tangential wavevector along the interface, the incident spatial frequency component with $(k_x, k_y)$ in the incident plane only generates the reflected spatial frequency component with the same $(k_x, k_y)$ in the output plane. At each $(k_x, k_y)$, the reflected wave has an electric field

$$\widetilde{\mathbf{E}}_{\mathbf{ref}}(k_x, k_y) = \mathbf{R}(k_x, k_y) \cdot \mathbf{e_{in}} \tilde{S}_{in}(k_x, k_y), \quad (2)$$

where $\mathbf{R}(k_x, k_y)$ is a $2 \times 2$ matrix. In general, the reflected waves can have different polarizations at different wavevectors. The reflected beam passes through a polarizer selecting an output polarization $\mathbf{e_{out}}$, resulting in an output electric field $\mathbf{E_{out}} = \mathbf{e_{out}} S_{out}(x,y)$, where $S_{out}(x,y)$ is a scalar field that describes the field distribution on the output plane, which is perpendicular to the propagation direction of the reflected beam. Here, we note that the coordinates $x$ and $y$ for $S_{in}(x,y)$ and $S_{out}(x,y)$ are defined in the reference planes for the incident and reflected beams, respectively. Similar to Eq. (1), we denote $\tilde{S}_{out}(k_x, k_y)$ as the spatial Fourier spectrum of $S_{out}(x,y)$. The entire reflection process is then described by $\tilde{S}_{out}(k_x, k_y) = r(k_x, k_y)\tilde{S}_{in}(k_x, k_y)$, where the scalar transfer function $r(k_x, k_y) = \mathbf{e_{out}^{\dagger}}\mathbf{R}(k_x, k_y)\mathbf{e_{in}}$. In this case, based on the reflection-induced $k$-dependent polarization variation, our setup performs nonlocal spatial filtering.

In order to achieve edge detection, we require that $S_{out}(x,y) = \hat{D}S_{in}(x,y)$, where $\hat{D}$ is a differential operator. For example, $\hat{D}$ can be a Laplacian[38], $\hat{D} = \partial_x^2 + \partial_y^2$, corresponding to a transfer function $r(k_x, k_y) = k_x^2 + k_y^2$ in the wavevector space. However, a wide variety of other differential operators can also be used for edge detection purposes. By definition, in order to achieve differential operations, the transfer functions has to have $r(k_x = 0, k_y = 0) = 0$. (Here $k_x = k_y = 0$ corresponds to the direction of the beam propagation, with an incident angle $\theta_0$.) In our setup, one can straightforwardly achieve $r(k_x = 0, k_y = 0) = 0$ by choosing the appropriate input and output polarizations such that

$$\mathbf{e_{out}^{\dagger}}\mathbf{R}(k_x = 0, k_y = 0)\mathbf{e_{in}} = 0. \quad (3)$$

Below we refer to the condition of Eq. (3) as the cross-polarization condition. When this condition is satisfied, the transfer function in the vicinity of $k_x = k_y = 0$ has the generic form (Here, we use similar notations following those in ref. [34], see detailed derivation in the Supplementary Note 1.):

$$r(k_x, k_y) = C_x k_x + C_y k_y, \tag{4}$$

as can be obtained from Taylor expansion of $r(k_x, k_y)$ to its first-order terms near $k_x = k_y = 0$. Since $r(k_x, k_y)$ is a complex scalar field, every loop $c$ in the $k_x - k_y$ space is associated with a winding number $l = \frac{1}{2\pi i} \oint_c \frac{dr}{r}$, which can only take integer values. $l \neq 0$ implies that the loop $c$ encloses at least one zero of $r(k_x, k_y)$. Therefore, for every isolated zero of $r(k_x, k_y)$, one can define its topological charge as the winding number of a loop that encloses only this particular zero. Since the winding number is an integer, it is topologically invariant. Any small continuous deformation of the system cannot change the topological charge of the zero. Suppose $C_y/C_x = Ae^{i\varphi}$, where $A \in R$ and $\varphi \in [0, \pi)$, the zero of $r(k_x, k_y)$ has a topological charge $l = \text{sgn}(A\varphi)$.

Moreover, in order to achieve two-dimensional isotropic differentiation, the transfer function must have a rotationally invariant magnitude, and hence the coefficients $C_x$ and $C_y$ have to satisfy

$$\frac{C_y}{C_x} = \pm i. \tag{5}$$

The transfer function then corresponds to a differential operator

$$\hat{D} = C_x(-i\partial_x \pm \partial_y). \tag{6}$$

For a given input distribution $S_{in}(x, y)$, this operator results in the following output intensity distribution:

$$I_{out}(x, y) \propto \left|\frac{\partial S_{in}}{\partial x}\right|^2 + \left|\frac{\partial S_{in}}{\partial y}\right|^2 \equiv |\nabla S_{in}|^2. \tag{7}$$

Thus, this operator nonlocally calculates the squared gradient magnitude of the incident field and can be employed for isotropic edge detection in 2D images. We note that with Eq. (5), the transfer function has the form $r(k_x, k_y) \propto e^{\pm i\vartheta}$, where $\tan \vartheta = k_y/k_x$, which shows that the zero of $r(k_x, k_y)$ has a topological charge of ±1. In contrast, the topologically trivial case, where $C_x$ and $C_y$ have the same phases, does not support the rotationally invariant response and can only perform one-dimensional differentiation along certain orientation[34]. Therefore, there is a connection between the topological charge of the transfer function, and the operation of edge detection. This connection has not been previously recognized. All previous works[14–39] on spatial differentiation utilize a transfer function that has no topological charge.

For ease of experimental implementation, we choose the output polarization to be a linear polarization with a normalized electric field $(-\sin \gamma, \cos \gamma)^T$, as schematically shown in Fig. 1. Based on Eq. (3), the corresponding incident polarization $\mathbf{e_{in}}$ can be determined as

$$\mathbf{e_{in}} = N \begin{pmatrix} r_{s0} \cos \gamma \\ -r_{p0} \sin \gamma \end{pmatrix}. \tag{8}$$

Here, $r_{p0}$ and $r_{s0}$ are the Fresnel reflection coefficients for p- and s-polarized plane waves at incident angle $\theta_0$, respectively, while $N = 1/\sqrt{|r_{s0} \cos \gamma|^2 + |r_{p0} \sin \gamma|^2}$ is a normalization factor. $C_x$ and $C_y$ can thus be calculated as (see derivation in the Supplementary Note 1)

$$C_x = N \frac{\sin \gamma \cos \gamma}{k_0} \left( r_{s0} \frac{\partial r_p}{\partial \theta} - r_{p0} \frac{\partial r_s}{\partial \theta} \right), \tag{9a}$$

$$C_y = -N \frac{\cot \theta_0}{k_0} (r_{p0} \sin^2 \gamma + r_{s0} \cos^2 \gamma)(r_{p0} + r_{s0}), \tag{9b}$$

where $\partial r_{p(s)}/\partial \theta$ is the first-order derivative of the Fresnel reflection coefficient for p(s)-polarized plane wave with respect to the incident angle ($\theta$) at $\theta_0$. To achieve an isotropic differentiation operation, for a given reflector, we combine Eqs. (9a) and (9b) with Eq. (5) to determine the required $\gamma$ and $\theta_0$. Once these two parameters are determined, the input polarization is determined using Eq. (8). In this way, we can determine the parameters required for an isotropic edge detection.

We note that in order to achieve a nontrivial topological charge in $r(k_x, k_y)$, $r_p$ and $r_s$ cannot be both real, since in this case $C_x$ and $C_y$ are also real numbers as can be seen in Eqs. (9a) and (9b). The nontrivial topological charge can be achieved with a wide variety of reflectors, for example lossy metallic reflectors, using photonic band gap effects, or using total internal reflection at a dielectric interface, since in all these cases $r_p$ and $r_s$ are complex with different phases. At a dielectric interface, below the critical angle of total internal reflection, the partial reflection on a dielectric interface cannot exhibit the topological charge in most cases since both $r_p$ and $r_s$ are real, except if one of the Fresnel reflection coefficients is zero which occurs at the Brewster angle where $r_{p0} = 0$. In this case, the cross-polarization condition [Eq. (3)] can be satisfied with an arbitrary input polarization $\mathbf{e_{in}} = (e_{in}^x, e_{in}^y)^T$. By setting $\gamma = \pi/2$, Eqs. (9a) and (9b) then become

$$C_x = \frac{1}{k_0} \frac{\partial r_p}{\partial \theta} e_{in}^x, \tag{10a}$$

$$C_y = \frac{\cot \theta_0}{k_0} r_{s0} e_{in}^y. \tag{10b}$$

With either a circular or an elliptical input polarization, $r(k_x, k_y)$ exhibits a nontrivial topological charge and enables two-dimensional differentiation, which, however, is anisotropic in most cases. Also, Eq. (5), which is required for isotropic differentiation, can be achieved with a correct choice of the input polarization. In this paper, we experimentally demonstrate isotropic edge detection by operating either with the total internal reflection (Fig. 2a) or at the Brewster angle (Fig. 2h).

**Measurement of the transfer functions and topological charges.** For the total internal reflection configuration, light beam is incident from the glass side of an air-glass interface onto the base of a prism. Based on Eqs. (5), (9a) and (9b), and using light at the wavelength of 532 nm, we determine the angle of incidence $\theta_0 = 70.24°$. In this case, we use an isosceles prism with 70.24° base angles (Fig. 2a) and send in light from air through the lateral side at normal incidence, such that the Goos-Hänchen and Imbert-Fedorov effects during the transmission can be neglected[41]. Specifically, we launch a Gaussian light beam from a laser at 532 nm through a linear polarizer and a subsequent quarter-wave plate to shape the input polarization, and select the output polarization with a second linear polarizer. Then, we acquire the magnitude squared of the output spatial spectrum by performing optical Fourier transform on the reflected light, which is measured by a CMOS camera (see setups in the Supplementary Note 2). By normalizing the magnitude of the output spatial spectrum with that of the incident one, we obtain the magnitude of the transfer function, i.e. $|r(k_x, k_y)|$.

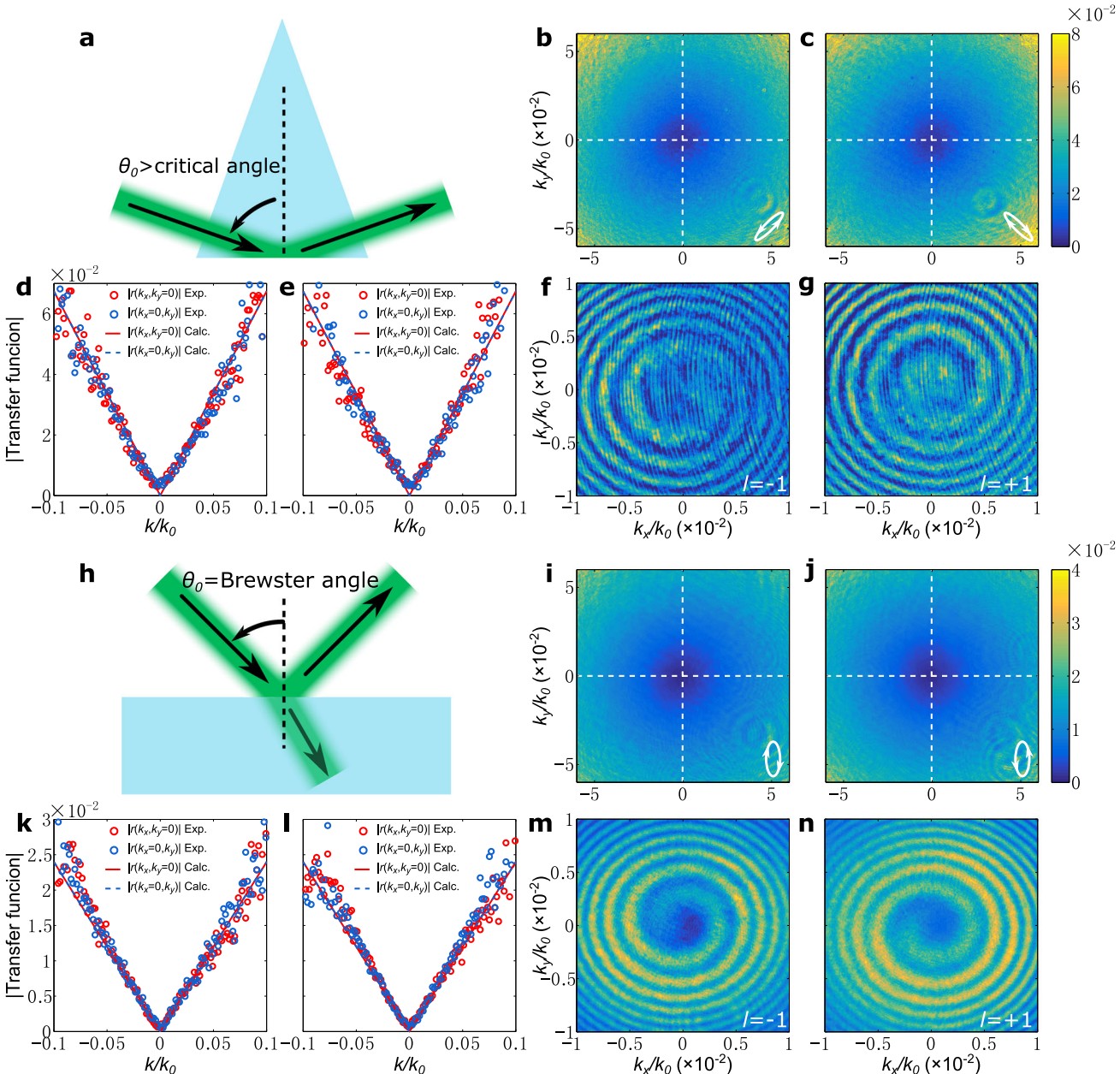

**Fig. 2 Measured magnitudes of transfer functions and corresponding topological charges. a** Schematic of the total internal reflection setup with $\theta_0 = 70.24°$. **b, c** Measured magnitudes of transfer functions, as a function of transverse wavevectors, using the setup in (**a**) with $\gamma = -\pi/4$ and $\gamma = +\pi/4$, respectively. The corresponding incident polarizations are shown as the insets. **d, e** The scatter plots are the data on the dashed lines in (**b, c**), respectively, and the lines are the corresponding calculated magnitudes of transfer functions. $k$ in the horizontal axes is the magnitude of the wavevector. **f, g** The interference fringe patterns of an output beam with a divergent beam corresponding to (**b, c**), respectively. The insets show the topological charges. **h** Schematic of the setup for operating at the Brewster angle with $\theta_0 = 56.64°$ and $\gamma = \pi/2$. **i, j** Measured magnitudes of transfer functions, as a function of transverse wavevectors, using the setup in (**h**) with the input polarizations shown as the insets. **k, l** The scatter plots are the data on the dashed lines in (**i, j**), respectively, and the lines are the corresponding calculated magnitudes of transfer functions. $k$ in the horizontal axes is the magnitude of the wavevector. **m, n** The interference fringe patterns of an output beam with a divergent beam corresponding to (**i, j**), respectively. The insets show the topological charges.

Figure 2b, c are the measured magnitudes of the transfer functions. Specifically, Fig. 2b is measured with the output polarizer orientation $\gamma = -\pi/4$ and the input polarization $\mathbf{e_{in}} = (r_{s0}, r_{p0})^T/\sqrt{2}$, as determined using Eqs. (5), (9a) and (9b) for the case of $C_y/C_x = -i$. Figure 2c is measured with output polarization orientation $\gamma = +\pi/4$ and incident polarization $\mathbf{e_{in}} = (r_{s0}, -r_{p0})^T/\sqrt{2}$, corresponding to the case of $C_y/C_x = +i$ in Eqs. (9a) and (9b). In both Fig. 2b, c, the measured magnitudes

of transfer functions are isotropic in the $k_x$–$k_y$ plane, and exhibit isolated zeros at $k_x = k_y = 0$.

To demonstrate the topological charges associated with the isolated zeros, based on the setup for measuring the transfer functions, we use an additional divergent reference beam to interfere with the output beam (see setups in the Supplementary Note 2). Corresponding to the cases shown in Fig. 2b, c, respectively, the measured interference patterns in Fig. 2f, g indeed show spiral fringes. The clockwise (Fig. 2f) and

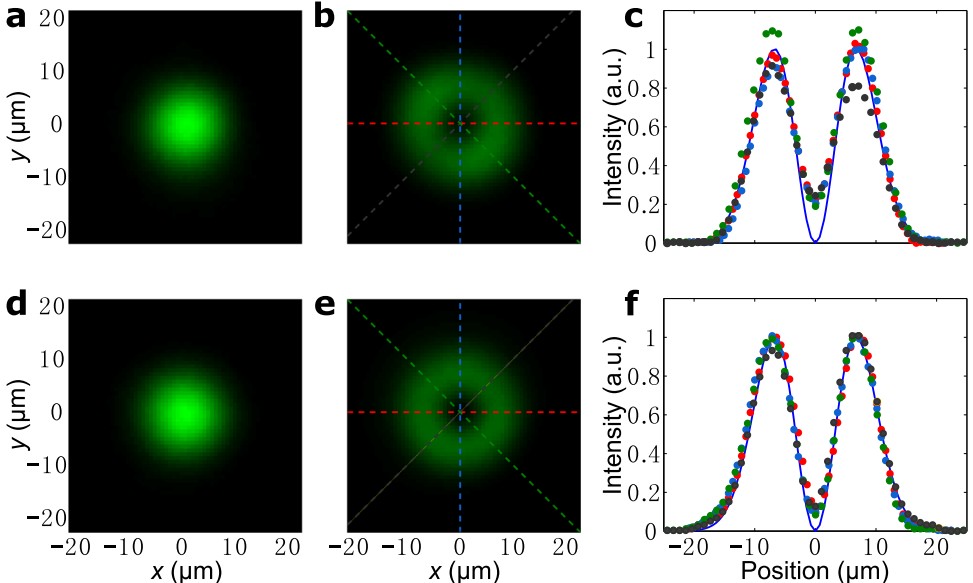

**Fig. 3 Spatial differentiation on an input Gaussian beam and generation of orbital angular momentum. a** Incident Gaussian beam using the total internal reflection setup shown in Fig. 2a. **b** Measured output beam, corresponding to the transfer function in Fig. 2b. **c** Profiles of the output beam along different directions. The dotted lines correspond to the data on the same-color dashed lines in (**b**), and the solid line is the ideal result, as calculated by using Eq. (6) to operate on an input Gaussian beam. **d–f** Same as (**a-c**), except for that we operate at the Brewster angle in a setup as shown in Fig. 2h. In **e** the used transfer function is the one shown in Fig. 2i.

counterclockwise (Fig. 2g) rotating directions of the spiral fringes indicate the opposite topological charges of $l = -1$ and $l = +1$, respectively.

As a quantitative evaluation of the isotropic behavior, we plot the $|r(k_x, k_y)|$ along the $k_x = 0$ and $k_y = 0$ lines as a function of wavevector in Fig. 2d, e, corresponding to Fig. 2b, c, respectively. $|r(k_x, k_y)|$ shows linear dependency as a function of magnitude of the wavevector, in agreement with Eq. (4). Moreover, the slopes of $|r(k_x, k_y)|$ along the two lines agree with each other, as well as with the theoretical results as calculated using Eqs. (9a) and (9b).

Figure 2i–n show the results when we operate at the Brewster angle. Figure 2h shows the schematic of the setup, where light is incident from the air side onto the air-glass interface. We set $\gamma = \pi/2$ and choose the input polarizations based on Eqs. (5), (10a) and (10b) for both cases of $C_y/C_x = -i$ and $C_y/C_x = +i$. Similar to the total internal reflection case, the magnitudes of the resulting transfer functions are isotropic in the $k_x$–$k_y$ plane with the isolated zeros at $k_x = k_y = 0$ (Fig. 2i–l). There are topological charges associated with the isolated zeros, as evidenced by the spiral interference patterns shown in Fig. 2m, n. The results in Fig. 2 indicate that we can indeed achieve the transfer function of Eq. (4), using either total internal reflection or by operating at the Brewster angle.

**Demonstration of isotropic differentiation and edge detection.** We now experimentally demonstrate isotropic spatial differentiation using the transfer functions in Fig. 2. We focus on the cases with topological charges of $-1$. The effects from the cases with topological charges of $+1$ are similar. The transfer functions with topological charge $-1$ are shown in Fig. 2b when we use total internal reflection, and in Fig. 2i when we operate at the Brewster angle. Mathematically, the transfer functions should correspond to the differential operator in Eq. (6) with a minus sign.

We first consider the scenario where the input, as shown in Fig. 3a, d, are Gaussian beams. The operator in Eq. (6) with a minus sign, operating on a Gaussian beam, should produce a beam with an orbital angular momentum of $-1$, which possesses

a donut-shaped intensity profile. We indeed observe such intensity profiles in the output beams, when we either use total internal reflection (Fig. 3b) or operate at the Brewster angle (Fig. 3e). In Fig. 3c, we show the intensity plots of the output beam as a function of position, along various directions that pass through the center of beam, for the case where we use total internal reflection. These plots strongly overlap with one another, indicating the isotropic differentiation. Similar isotropic behavior can be seen when we operate at the Brewster angle (Fig. 3f). For comparison, we also calculated and plot the ideal squared gradient magnitude of a Gaussian beam in both Fig. 3c, f. The experimental results coincide well with the calculated ones, indicating a great performance of the differentiation.

We now demonstrate the isotropic edge enhancement by directly incorporating our setup as part of an imaging system. Different from the Laplacians[38,39], the proposed first-order differentiator is able to directly detect edges in images without the need for subsequent zero-crossing detection. Here we focus on the case where we use total internal reflection. The results from operating at the Brewster angle are very similar and not shown here. We first send a collimated laser beam through a binary mask and then use an imaging system to image the mask onto a CMOS camera (Fig. 4a). Figure 4b shows the input image of a Stanford logo, when we use a laser at the wavelength of 532 nm. Since our differentiator works as a nonlocal filter, it can be placed at any position in the object or image space. As shown in Fig. 4a, we incorporate the differentiator in the imaging system and thus have the input image differentiated (see details in the Supplementary Note 3). The corresponding output image is shown in Fig. 4c. We see that the output image consists of only the edges, and all edges appear in the image with similar magnitudes independent of their orientations. The image here provides a clear demonstration of the isotropic edge enhancement. Since we propose our design in the paraxial regime, it has a finite spatial bandwidth, which means it cannot resolve edges that are too close. In Fig. 4d, we show the output image when the input one consists of an array of full circles with gradually reducing diameters. The output image consists of circumferences

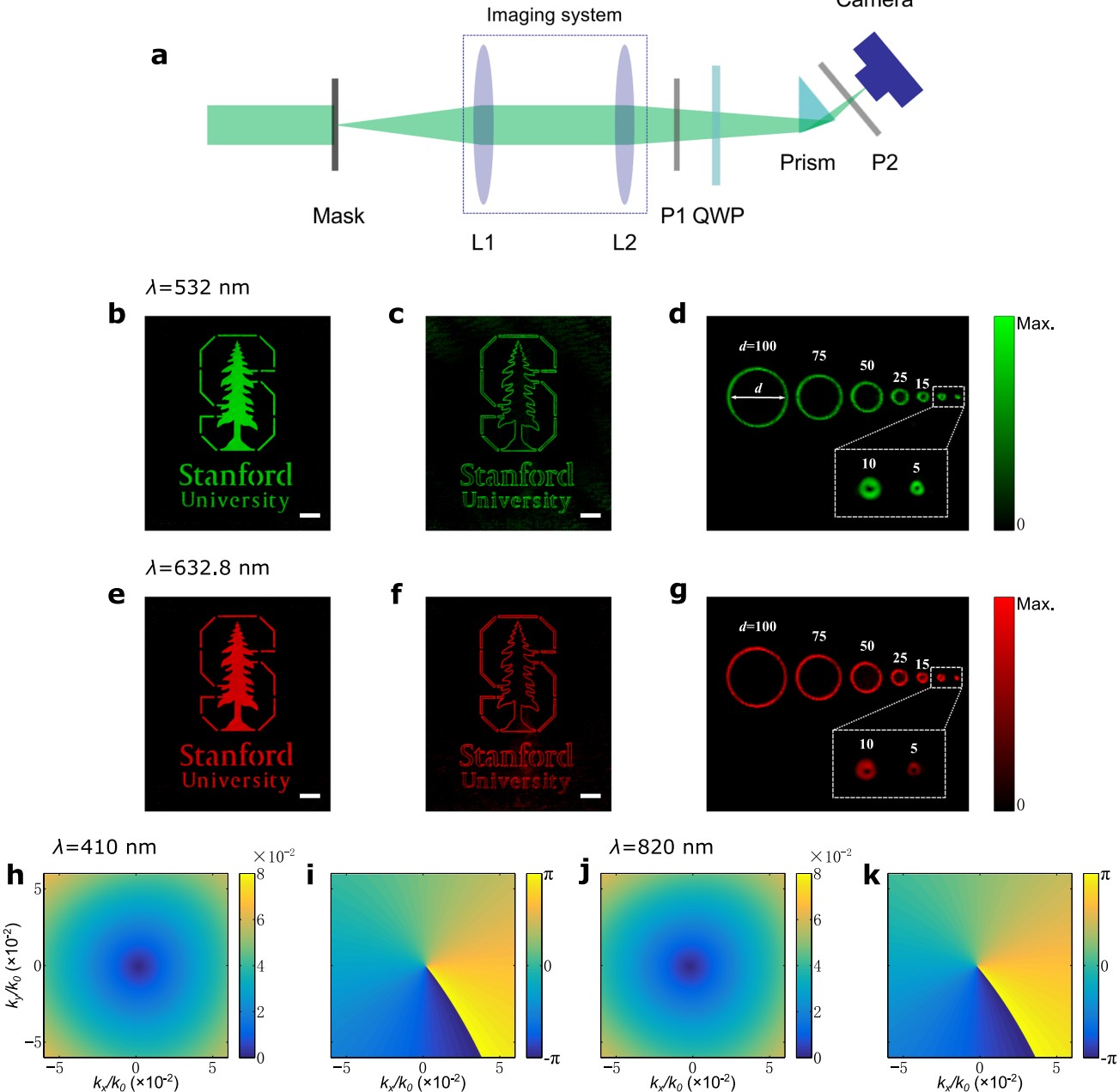

**Fig. 4 Demonstration of isotropic edge detection at different wavelengths. a** Experimental setups for edge detection when using total internal reflection. L1 and L2, lenses; P1 and P2, linear polarizers; QWP, quarter-wave plate. **b** Incident image of Stanford logo with the illumination wavelength of 532 nm. **c** Measured image corresponding to (**b**) using setup shown in (**a**). **d** Resolution test results, corresponding to an input image consisting of an array of full circles with gradually reducing diameters. **e**–**g** Same as (**b**-**d**), except that we operate at the wavelength of 632.8 nm. **h**, **i** Numerically calculated magnitudes and phases of the simulated transfer function at the wavelength of 410 nm, respectively. **j**, **k** Same as (**h**, **i**), except that we operate at the wavelength of 820 nm. The white bars correspond to a length of 200 μm. *d* in (**d**, **g**) denotes the diameters of incident full circle patterns in μm.

corresponding to the boundaries of the circles (see details in Supplementary Note 3). The intensity on the circular edges is quite uniform, which again points to the isotropic nature of the differentiation operation. The edges are clearly visible even for an input full circle with a diameter as small as 5 μm. Thus, the ability of the differentiation operation to resolve close edges here should be sufficient for most image processing applications.

In contrast to all previous works on using nanophotonic structures for spatial differentiation, in our scheme, the zero of the transfer function carries a topological charge. Hence the presence of zero cannot be removed by small perturbations. The

only mechanisms to eliminate a zero are to merge two zeros with opposite charges so that they annihilate each other, or to move an isolated zero outside the light cone so that it becomes inaccessible from free space. As long as the isolated zero persists within the light cone, 2D differentiation can be achieved by aligning the incident wavevector with the wavevector at which the transfer function $r$ vanishes. Therefore, there is a certain level of robustness of our device with respect to parameter variations. The level of robustness is defined by the strength of system variations that is required to eliminate a zero through the mechanisms mentioned above. We note that the robustness of the

zero in the transfer function is a topological effect that does not require the system to have any symmetry.

Moreover, a highly important characteristic is that with the same experimental parameters, the differentiator can operate over a broad spectral band. Here the broadband nature also arises from the fact that the refractive index of glass does not vary significantly as a function of frequency. As an illustration of the broad spectral bandwidth, for the total internal reflection case, in Fig. 4h, i, we numerically calculate and show the magnitude and phase of the transfer function, respectively, at the wavelength of 410 nm. And in Fig. 4j, k, we show the same at the wavelength of 820 nm. In these plots, except the wavelengths, all other operating conditions, including the angle of incidence, the polarization of the incident light, and the orientation of the output polarizer, are kept the same. In spite of a change of factor 2 in operating frequencies, the normalized transfer functions in the two cases are very similar to each other. Here the small difference between the two arises from the small differences in the refractive index of glass at these two wavelengths. We note that the transfer functions at different wavelengths have different slopes with respect to $k_x$ and $k_y$, thus the strength of the differentiation result is wavelength-dependent. Such wavelength dependency, however, can be easily calibrated in multispectral applications.

The above discussion indicates that this device can operate over a broad spectral band. As a direct illustration, Fig. 4e–g show the experimental results as Fig. 4b–d, except that we now use a laser at the wavelength of 632.8 nm, which has a red color. We observe near-identical performances at these two wavelengths. The operating spectral bandwidth here far exceeds all previous works on using photonic structures for two-dimensional spatial differentiation.

## Discussion

In conclusion, we have established a connection between two-dimensional optical spatial differentiation and a nontrivial topological charge in the optical transfer function. Based on this connection, we experimentally demonstrate broadband isotropic two-dimensional differentiation, by generating a nonzero topological charge with a cross-polarization setup of light reflection at an unpatterned interface. The nontrivial topological charge plays an essential role for the two-dimensional operation, since the topologically trivial case can only perform one-dimensional directional differentiation. We note that although we choose a linear output polarization for the ease of experiment, circular or elliptical output polarizations can also be used for the isotropic differentiation, and there is no longer the restriction of using total internal reflection or operating at the Brewster angle in these cases. We anticipate that similar setup can be used to generate nontrivial topological charges for other configurations of photonic structures. Moreover, it should be possible to use photonic structures in such setup to generally engineer nonlocal responses for other computational tasks. For example, higher-order differentiation may be achieved by generating multiple topological charges in the transfer function. We anticipate that exploitation of optical transfer functions with different nontrivial topological charge configurations can lead to new opportunities in achieving general and robust optical computing.

## Data availability

The data that support the findings of this study are available from the corresponding author upon reasonable request.

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

## Acknowledgements
We thank Dr. Kai Wang for discussions. Z.R., T.Z., and J.H. acknowledge National Key Research and Development Program of China (Grant No.2017YFA0205700), National Natural Science Foundation of China (NSFC Grants No. 91850108 and No. 61675179), the Open Foundation of the State Key Laboratory of Modern Optical Instrumentation, and the Open Research program of Key Laboratory of 3D Micro/Nano Fabrication and Characterization of Zhejiang Province. This work is supported by a MURI project from the U.S. Air Force Office of Scientific Research (AFOSR) Grant No. FA9550-17-1-0002, by a U.S. Office of Naval Research (ONR) Grant No. N00014-20-1-2450 and by a Vannevar Bush Faculty Fellowship from the U. S. Department of Defense (Grant No. N00014-17-1-3030). T.Z. acknowledges the support by the China Scholarship Council (Scholarship 201906320169).

## Author contributions
T.Z. conceived the idea. T.Z., J.H., and H.W. carried out the experiments. S.F., T.Z., C.G., and M.O. wrote the paper with input from all the authors. S.F. and Z.R. jointly supervised this work.

## Competing interests
The authors declare no competing interests.
