## [Peer Review File · Nature Communications]

Reviewers' Comments:

Reviewer #1:

Remarks to the Author:

The authors have proposed a broadband two-dimensional spatial differentiator, and establish a connection between this operation and topological physics. I found the idea proposed in the manuscript intriguing. The authors have presented the results quite clearly. The obtained results look sound. The topic of the paper covers a broad range of fields of interest, including topological physics, metamaterials, and analog signal processing. As a result, I would like to recommend the paper for publication in Nature Communications. Yet, there are a couple of issues that have to be addressed by the authors.

1- The authors argue that the proposed 2D differentiation operator is associated with a topological charge. However, they do not provide any evidence of the topological origin of this number. In particular, I am wondering what protects the topological invariant proposed by the authors? Are the authors sure that this number cannot be changed unless the system goes through a discontinuous deformation? How is this quantity protected?

2- The authors state that "there is certain level of robustness of our device with respect to parameter variations". What is this certain level? For ordinary topological insulators, this certain level is related to the so-called Anderson localization. Do we have something similar here?

3- Also, regular topological insulators are robust only against certain classes of disorder, depending on the symmetry that protects the invariant. Is the robustness of the proposed topological invariant also limited to a specific type of disorder?

4- The authors state that "Our work indicates that exploiting the concepts in topological photonics can lead to new opportunities in optical computing." However, this has already been demonstrated in Ref. 11 of the paper. I believe this reference deserves a better citation and discussion in the main text (introduction).

Minor points:

The color bars in Fig. 4 are missing.

There are a few language glitches in the text such as "there is certain level of...". Please proofread the article.

Reviewer #2:

Remarks to the Author:

The manuscript titled "Topological optical differentiator" by T. Zhu et al. studies a possible connection between optical transfer function in form of spatial differentiation and presence of nontrivial topological charge in the transfer function. The authors consider a simple and easily available platform (dielectric interface with no patterning) throughout the text to demonstrate their findings. The manuscript is quite well-written, has proper length, and figures are clear and illustrative. I have several comments/questions which I listed below and not necessarily in the order of importance:

(A) The two-dimensional transfer function of the interface takes a generic linear form following similar notation/formulation as given in Ref. [34], calculated in eqs. (4)-(9). While the calculations are necessary to make the current manuscript an easy and independent read, it would be appropriate to better refer to [34] regarding the notation and formulations used here.

(B) The goal "functionality" in this manuscript is 2D and isotropic differentiation which naturally

requires a zero at $k_x=k_y=0$. Therefore, the discussions around presence (or absence) of a nontrivial topological charge in the transfer function are limited to this specific functionality. I believe that the general conclusion of the manuscript "exploiting the concepts in topological photonics can lead to new opportunities in optical computing" is insightful and reasonable but is not necessarily a direct continuation of this work. Considering the scope/aim of the manuscript there is plenty of room for discussion about such possibilities in the conclusion and I encourage the authors to better discuss this point. Can this concept be useful to achieve robust and "general" optical transfer functions?

(C) The condition in eq. (5) is a key element in the design as it distinguishes between isotropic and anisotropic 2D differentiation. The extreme later case is a 1D differentiator and the isotropic case is the one studied here. Considering the rather stringent requirements on the relation between the polarization/angle of the incident and reflected waves (discussions after eq. (10) and supplementary materials), is it possible to create a tradeoff between the level of anisotropy in the transfer function and the polarization relation in the input and output waves? In other words, can we create an anisotropic differentiator that works for any polarization or angle of incidence?

(D) The authors state that "All previous works [14-39] on spatial differentiation utilize a transfer function that has no topological charge." Researchers have used several techniques ranging from optimization, coupling to resonant modes/phenomena, relying on symmetries/asymmetries in the physical properties of a (patterned) surface, etc. to achieve spatial differentiation. While I am not aware of a previous research on using topological charges to design optical differentiators, how can the authors be sure that the previous successful proposals did not accidentally use such an effect? Can the authors comment on this?

(E) What does determine the resolution in this design? i.e. the 5 μ m resolution in Fig. 4.

(F) Can you comment more on the relations between the presented structure and the one presented in [39]? Is the accessible bandwidth the only difference from a user point of view?

(G) While the response of the system is clearly broadband, I am not convinced that this is a specifically engineered feature. As stated in the text: "The zero of the transfer function carries a topological charge. Hence the presence of zero cannot be removed by small perturbations. As a result, there is certain level of robustness of our device with respect to parameter variations." But does this justify operation at twice the original wavelength? If a plasmonic setup has been used, would you expect similar broadband response?

(H) Can you comment on possible relations with the results presented in [32] in terms of operation bandwidth? Following on point (D), is it possible that some previous works have accidentally utilized a topological charge in transfer function?

Overall, I believe this is a relevant and well-executed research with a novel idea and the subject matter is suitable for Nature Communications. I recommend a revision at this point and upon successful response to the above comments I believe this manuscript can be suitable for publication in Nature Communications.

Point-by-Point Response to Referee #1's Comments:

To Comment (1): “The authors argue that the proposed 2D differentiation operator is associated with a topological charge. However, they do not provide any evidence of the topological origin of this number. In particular, I am wondering what protects the topological invariant proposed by the authors? Are the authors sure that this number cannot be changed unless the system goes through a discontinuous deformation? How is this quantity protected?”

Response: Here the topological charge is the winding number of the complex transfer function around an isolated complex zero. Specifically, since the transfer function $r(k_x, k_y)$ is a complex scalar field, one can compute the winding number l of $r(k_x, k_y)$ along a loop c in the $k_x - k_y$ space:

$$l = \frac{1}{2\pi i} \oint_c \frac{dr}{r}.$$

l can only take integer values and moreover can only take non-zero values if the loop c encloses a zero of r . In our case, $r(k_x, k_y) = C_x(k_x \pm ik_y)$ thus $l = \pm 1$, when the loop is chosen to encircle the origin of $k_x = k_y = 0$ where $r = 0$. Since l is constrained to be an integer, it is topologically invariant: a small continuous deformation of the system cannot change the value of such an integer. Moreover, a non-zero l for a loop guarantees that the loop always encircles a wavevector at which $r = 0$, which is essential for our purpose of 2D differentiation.

Modification: To make these points clear, we have added the following sentences:

(the 70th line in Page 3) “Since $r(k_x, k_y)$ is a complex scalar field, every loop c in the $k_x - k_y$ space is associated with a winding number $l = \frac{1}{2\pi i} \oint_c \frac{dr}{r}$, which can only take integer values. $l \neq 0$ implies that the loop c encloses at least one zero of $r(k_x, k_y)$. Therefore, for every isolated zero of $r(k_x, k_y)$, one can define its topological charge as the winding number of a loop that encloses only this particular zero. Since the winding number is an integer, it is topologically invariant. Any small continuous deformation of the system cannot change the topological charge of the zero.”

(the 82nd line in Page 3) “We note that with Eq. (5), the transfer function has the form $r(k_x, k_y) \propto e^{\pm i\vartheta}$, where $\tan \vartheta = k_y/k_x$, which shows that the zero of $r(k_x, k_y)$ has a topological charge of ± 1 .”

To Comment (2): “The authors state that “there is certain level of robustness of our device with respect to parameter variations.” What is this certain level? For ordinary topological insulators, this certain level is related to the so-called Anderson localization. Do we have something similar here?”

Response: We thank the referee for this insightful comment. By certain level we mean the system variation is not strong enough to eliminate the isolated zero of $r(k_x, k_y)$ in the wavevector range within light cone. As we mentioned above, such an isolated zero has an associated winding number $l = \pm 1$ and thus cannot be eliminated by small variations of the system. The only mechanisms to eliminate a zero are to merge two zeros with opposite charges so that they annihilate each other, or to move an isolated zero outside the light cone so that it becomes inaccessible from free space. As long as

the isolated zero persists within the light cone, 2D differentiation can be achieved by aligning the incident wavevector with the wavevector at which the transfer function r vanishes. Therefore, the strength of system variations that is required to eliminate a zero through these mechanisms defines the level of robustness.

Our work concerns the topology of zeros in the transfer function. Regarding relations to topological materials, the closest analogy with what we do here is the topology of Weyl semimetal. The Weyl node in a Weyl semimetal, which is protected by an integer topological charge, also can only be eliminated by merging two nodes with opposite charges together, similar to the merging of the zeros that we are discussing here.

Modification: To better clarify these points, we add the following explanations:

(the 186th line in Page 7) “Hence the presence of zero cannot be removed by small perturbations. The only mechanisms to eliminate a zero are to merge two zeros with opposite charges so that they annihilate each other, or to move an isolated zero outside the light cone so that it becomes inaccessible from free space. As long as the isolated zero persists within the light cone, 2D differentiation can be achieved by aligning the incident wavevector with the wavevector at which the transfer function r vanishes. Therefore, there is a certain level of robustness of our device with respect to parameter variations. The level of robustness is defined by the strength of system variations that is required to eliminate a zero through the mechanisms mentioned above.”

To Comment (3): “*Also, regular topological insulators are robust only against certain classes of disorder, depending on the symmetry that protects the invariant. Is the robustness of the proposed topological invariant also limited to a specific type of disorder?*”

Response: The referee is indeed correct that in regular topological insulators, the topological phase is protected by time-reversal symmetry, and the system is robust only against disorders that preserve time-reversal symmetry. However, not all topological phases require such symmetry-related protection. For example, the Weyl node in a Weyl semimetal, as we discussed above, is protected by topology alone. The existence of Weyl node does not require the presence of any symmetry. (In fact, it requires the *absence* of either time-reversal symmetry or spatial-inverse symmetry). Analogous to the Weyl node, the topological nature of the zero in the transfer function also does not require any symmetry of the system.

Modification: To make this point clear, we add the following sentence:

(the 191st line in Page 7) “We note that the robustness of the zero in the transfer function is a topological effect that does not require the system to have any symmetry.”

To Comment (4): “*The authors state that “Our work indicates that exploiting the concepts in topological photonics can lead to new opportunities in optical computing.” However, this has already been demonstrated in Ref. 11 of the paper. I believe this reference deserves a better citation and discussion in the main text (introduction).*”

Response and modification: We agree with the referee that Ref. 11 deserves a better citation in the introduction. To make a better citation and show the differences between Ref. 11 and our work, we add the following sentence:

(the 32nd line in Page 1) “Different from the work which uses topological states for one-dimensional transfer functions [11], we directly construct a topological charge in two-dimensional transfer functions to engineer the functionality of device.”

To Comment (5): *“The color bars in Fig. 4 are missing. There are a few language glitches in the text such as “there is certain level of...” Please proofread the article.”*

Response and modification: We thank the referee’s notification. We have added the color bars in Fig. 4. Also, we have corrected the language glitches and done more proofreading to the manuscript.

Point-by-Point Response to Referee #2's Comments:

To Comment (1): *“The two-dimensional transfer function of the interface takes a generic linear form following similar notation/formulation as given in Ref. [34], calculated in eqs. (4)-(9). While the calculations are necessary to make the current manuscript an easy and independent read, it would be appropriate to better refer to [34] regarding the notation and formulations used here.”*

Response and Modification: We agree with the referee that the linear transfer function uses similar notations and formulation as those in Ref. [34]. To show this point, we have added the following sentences:

(the 68th line in Page 3) “Here, we use similar notations following those in Ref. [34], see detailed derivation in the Supplementary Material.”

(the 11th line in Page 1 of the Supplementary Material) “In the derivation, we use similar notations and formulations following Ref. [34] of the main text, where one special case of cross polarization is discussed.”

To Comment (2): *“The goal “functionality” in this manuscript is 2D and isotropic differentiation which naturally requires a zero at $k_x=k_y=0$. Therefore, the discussions around presence (or absence) of a nontrivial topological charge in the transfer function are limited to this specific functionality. I believe that the general conclusion of the manuscript “exploiting the concepts in topological photonics can lead to new opportunities in optical computing” is insightful and reasonable but is not necessarily a direct continuation of this work. Considering the scope/aim of the manuscript there is plenty of room for discussion about such possibilities in the conclusion and I encourage the authors to better discuss this point. Can this concept be useful to achieve robust and “general” optical transfer functions?”*

Response and Modification: We thank the referee for this suggestion. We have added the following discussion about possibility of using the proposed concept for other computational tasks.

(the 221st line in Page 7) “Moreover, it should be possible to use photonic structures in such setup to generally engineer nonlocal responses for other computational tasks, by generating multiple topological charges in the transfer function.”

To Comment (3): *“The condition in eq. (5) is a key element in the design as it distinguishes between isotropic and anisotropic 2D differentiation. The extreme later case is a 1D differentiator and the isotropic case is the one studied here. Considering the rather stringent requirements on the relation between the polarization/angle of the incident and reflected waves (discussions after eq. (10) and supplementary materials), is it possible to create a tradeoff between the level of anisotropy in the transfer function and the polarization relation in the input and output waves? In other words, can we create an anisotropic differentiator that works for any polarization or angle of*

incidence?”

Response: We thank the referee for this very insightful comment. In fact, there can be such anisotropic differentiator that works for any polarization.

As we mentioned in the main text, the differentiator should first satisfy the cross-polarization condition, i.e. $e_{out}^\dagger R(k_x = 0, k_y = 0)e_{in} = 0$. Then, if one wants to achieve differentiation with an arbitrary angle of incidence, the cross-polarization condition needs to be satisfied with an arbitrary diagonal matrix $R(k_x = 0, k_y = 0)$. As a result, the polarizations will be $e_{in} = \begin{pmatrix} 1 \\ 0 \end{pmatrix}$ and $e_{out} = \begin{pmatrix} 0 \\ 1 \end{pmatrix}$ [or $e_{in} = \begin{pmatrix} 0 \\ 1 \end{pmatrix}$ and $e_{out} = \begin{pmatrix} 1 \\ 0 \end{pmatrix}$], and the corresponding differentiation becomes one-dimensional, i.e. the extreme anisotropic case.

On the other hand, the anisotropic differentiator can work for any input polarization at the Brewster angle, because as discussed in the manuscript, the cross-polarization condition can always be satisfied in this case. Specifically, with a linear input polarization, the differentiation will be one-dimensional. And with a circular or an elliptical input polarization, the differentiation will be two-dimensional but anisotropic in most cases. Moreover, the level of anisotropy can be tuned by choosing different input polarizations.

Modification: To show that anisotropic two-dimensional can be achieved with less stringent requirements on polarizations, we have added the following sentence:

(the 111th line in Page 4) “With either a circular or an elliptical input polarization, $r(k_x, k_y)$ exhibits a nontrivial topological charge and enables two-dimensional differentiation, which however is anisotropic in most cases.”

To Comment (4): “*The authors state that “All previous works [14-39] on spatial differentiation utilize a transfer function that has no topological charge.” Researchers have used several techniques ranging from optimization, coupling to resonant modes/phenomena, relying on symmetries/asymmetries in the physical properties of a (patterned) surface, etc. to achieve spatial differentiation. While I am not aware of a previous research on using topological charges to design optical differentiators, how can the authors be sure that the previous successful proposals did not accidentally use such an effect? Can the authors comment on this?*”

Response: We thank the referee for this question. We do believe that the cited papers are sufficiently representative of the current state of the art in the literatures on differentiation. We have examined every one of these papers cited. None of them used a transfer function that has a topological charge.

Most of the cited work demonstrated one-dimensional differentiation. As we mentioned in the main text, the nonzero topological charge arises from an isolated zero in the transfer function. Therefore, one-dimensional differentiations cannot hold a nonzero

topological charge, since the zeros in those transfer functions form a line in the complex plane, and are therefore not isolated at discrete points.

Two-dimensional differentiations can have topological charges. However, most proposals, for example Refs. [25-28, 38, 39], use a second-order nonlocal response of $k_x^2 + k_y^2$ or $Ak_x^2 + Bk_y^2$, neither of them have topological charges. Although Ref. [28] shows a transfer function of a 2D first-order differentiation, the form of the transfer function shown in Ref. [28] does not have a topological charge either.

To Comment (5): *“What does determine the resolution in this design? i.e. the 5um resolution in Fig. 4.”*

Response: The resolution of the device in edge detection is determined by its spatial bandwidth. Specifically, the spatial bandwidth of a differentiator is related to the width of a range in wavevector space where its transfer function is approximately linear. As discussed in Refs. [20, 29], a broader spatial bandwidth can result in a smaller resolution. On the other hand, in experimental measurement, the resolution is also limited by the numerical aperture of the imaging system. Our results show that the proposed device can at least resolve two edges that are 5 μ m away from each other.

Modification: To clearly show this point, we have added the following sentence:

(the 177th line in Page 6) “Since we propose our design in the paraxial regime, it has a finite spatial bandwidth, which means it cannot resolve edges that are too close.”

To Comment (6): *“Can you comment more on the relations between the presented structure and the one presented in [39]? Is the accessible bandwidth the only difference from a user point of view?”*

Response: From a user’s point of view, the spectral bandwidth is the most important difference between these two proposals, but not the only one. For example, another difference is that our structure performs first-order differentiation, while the Laplacian differentiation in Ref. [39] is second-order due to the symmetry of their setup. In image processing, a first-order differentiator can directly extract the edges, while a Laplacian requires a subsequent zero-crossing detection to get the edges. However, on the other hand, a Laplacian can be more effective for slow variations than the first-order differentiation. So, these two designs can be suitable for different application scenarios.

Modification: To make this point, we have added the following sentence:

(the 167th line in Page 6) “Different from the Laplacians [38, 39], the proposed first-order differentiator is able to directly detect edges in images without the need for subsequent zero-crossing detection.”

To Comment (7): *“While the response of the system is clearly broadband, I am not*

convinced that this is a specifically engineered feature. As stated in the text: “The zero of the transfer function carries a topological charge. Hence the presence of zero cannot be removed by small perturbations. As a result, there is certain level of robustness of our device with respect to parameter variations.” But does this justify operation at twice the original wavelength? If a plasmonic setup has been used, would you expect similar broadband response?”

Response: We thank the referee for this insightful comment. Since our design does not rely upon the use of resonance, it is not sensitive to the wavelength. In our case, the variation induced by the change of factor 2 in operating frequencies is only in the refractive index of the glass. And our topological differentiator is robust against such small parameter variations, so that it can work well over such a broad spectral bandwidth.

However, in plasmonic setups, the parameters, for example the reflection coefficients, are much more sensitive to wavelength variation due to the use of resonance. As a result, the spectral bandwidth will become much narrower.

Modification: To better explain the origin of the broadband feature, we have added the following sentence:

(the 195th line in Page 7) “Here the broadband nature also arises from the fact that the refractive index of glass does not vary significantly as a function of frequency.”

To Comment (8): *“Can you comment on possible relations with the results presented in [32] in terms of operation bandwidth? Following on point (D), is it possible that some previous works have accidentally utilized a topological charge in transfer function?”*

Response: As mentioned in last response, the broad spectral bandwidth of our design is due to its robustness against variations in the refractive index induced by wavelength changes. In Ref. [32], their device has a broadband property simply because it works on the basis of a wavelength-independent effect, i.e. geometric phase of the nanostructure orientation.

However, as explained in our response to comment (4), this one-dimensional differentiation cannot hold a topological charge in its transfer function.

Reviewers' Comments:

Reviewer #1:

Remarks to the Author:

The authors have adequately addressed my previous comments. I recommend publication of the work at this stage.

Reviewer #2:

None

Point-by-Point Response to Referee #1's Comments:

To Comment (1): *“The authors have adequately addressed my previous comments. I recommend publication of the work at this stage.”*

Response: We thank the referee for his/her time and the recommendation.